# Patient-Derived, Drug-Resistant Colon Cancer Cells Evade Chemotherapeutic Drug Effects via the Induction of Epithelial-Mesenchymal Transition-Mediated Angiogenesis

**DOI:** 10.3390/ijms21207469

**Published:** 2020-10-10

**Authors:** Jin Hong Lim, Kyung Hwa Choi, Soo Young Kim, Cheong Soo Park, Seok-Mo Kim, Ki Cheong Park

**Affiliations:** 1Gangnam Severance Hospital, Department of Surgery Yonsei, University College of Medicine 211 Eonjuro, Gangnam-gu, Seoul 135-720, Korea; DOCTORJIN@yuhs.ac (J.H.L.); KIMSUY@yuhs.ac (S.Y.K.); CSPARK1@yuhs.ac (C.S.P.); 2Department of Surgery, Yonsei University College of Medicine, 50-1, Yonsei-ro, Seodaemun-gu, Seoul 120-752, Korea; 3Department of Urology, CHA Bundang Medical Center, CHA University, Seongnam 463-712, Korea; khchoi@cha.ac.kr; 4Renal Division, Brigham and Women’s Hospital, Department of Medicine, Harvard Medical School, Boston, MA 02115, USA; 5Thyroid Cancer Center, Gangnam Severance Hospital, Department of Surgery, Yonsei University College of Medicine, Seoul 120-752, Korea

**Keywords:** patient-derived colon cancer, drug resistance, oxaliplatin, lenvatinib, angiogenesis

## Abstract

Cancer cells can exhibit resistance to different anticancer drugs by acquiring enhanced anti-apoptotic potential, improved DNA injury resistance, diminished enzymatic inactivation, and enhanced permeability, allowing for cell survival. However, the genetic mechanisms for these effects are unknown. Therefore, in this study, we obtained drug-sensitive HT-29 cells (commercially) and drug-resistant cancer cells (derived from biochemically and histologically confirmed colon cancer patients) and performed microarray analysis to identify genetic differences. Cellular proliferation and other properties were determined after treatment with oxaliplatin, lenvatinib, or their combination. In vivo, tumor volume and other properties were examined using a mouse xenograft model. The oxaliplatin and lenvatinib cotreatment group showed more significant cell cycle arrest than the control group and groups treated with either agent alone. Oxaliplatin and lenvatinib cotreatment induced the most significant tumor shrinkage in the xenograft model. Drug-resistant and metastatic colon cancer cells evaded the anticancer drug effects via angiogenesis. These findings present a breakthrough strategy for treating drug-resistant cancer.

## 1. Introduction

Cancer contributes to a considerable number of deaths worldwide [1,2,3,4]. Globally, in 2018, cancer was responsible for 9.6 million deaths; from being the third major cause of death in 1990, it became the second major cause, after heart disease, in 2018 [4]. Although complete remission is not possible, there has been a noticeable reduction in disease progression rates and an increase in the overall median survival rates [5,6]. Despite these developments, the need for an effective anticancer drug remains unmet owing to drug resistance [7,8,9]. Approximately 90% of therapeutic failure is caused by drug resistance [10]. Cancer-related deaths have increased over the past few decades and are still on the rise [11,12]. The Human Genome Project has increased our understanding of genetic factors involved in cancer development and has allowed for better diagnosis, early detection, and therapy [13]. However, it is not sufficient to identify molecular targets, as they can be undruggable and undergo epigenetic post-modification. Targeted therapy has taken the center stage in anticancer drug development [14,15,16,17,18]. Yet, research is ongoing to find an effective drug. Cancer chemotherapy failure can be attributed to numerous causes [19,20,21], including the development of resistance to many different types of drugs. Cancer cells acquire drug resistance through multidrug resistance factors, cell death inhibition (anti-apoptosis induction), modification of drug metabolism and epigenetic and drug targets, and enhanced DNA repair and genetic mutations [21,22]. Of course, some genetic mechanisms have been reported and a number of molecular targets have been identified, such as survivin, bcl-2, MMR, and NER systems. However, details on the genetic mechanisms of acquiring drug resistance remain unclear.

In this study, we discovered that anticancer drug-resistant colon cancer cells were able to avoid destruction by drug-induced apoptosis and thereby survive to a greater extent than drug-sensitive colon cancer cells. Moreover, when administered together, oxaliplatin and lenvatinib synergistically caused potent tumor shrinkage; however, there may be other unknown processes to consider, which emphasizes the need for additional serious contemplation toward designing and executing more deductive studies.

## 2. Results

### 2.1. Characteristics of Colon Cancer Cell Lines in This Study

Drug-resistant and metastatic colon cancer cells were acquired from the patient specimens and were compared with established HT-29 human colon cancer cell lines (Figure 1A). YUMC-C1 and YUMC-C2 were obtained from colon cancer patients treated at Severance Hospital, Yonsei University College of Medicine, Seoul, Korea. Unfortunately, however, despite clinical trials, including oxaliplatin and radiation therapy, these colon cancer patients experienced therapy failure, cancer recurrence, and metastasis. The results of the gene expression microarray analysis, YUMC-C1 and YUMC-C2, patient-derived drug-resistant and metastatic colon cancer cells were significantly increased levels of cancer stem cell marker, CD133^high^/CD44^high^/CD24^low^ (Figure 1B). Moreover, metastatic genes and markers of stemness were markedly enhanced. The nuclear trans-localization of β-catenin is crucial for the EMT (epithelial-mesenchymal transition) that was significantly induced in YUMC-C1 and YUMC-C2 (Figure 1C), as opposed to HT-29. Furthermore, EMT-related genes were more significantly induced than HT-29 (Figure 1D). *APC*, *SMAD4*, *NF1* (neurofibromin 1), *ARID5B*, *NCOR1* (nuclear receptor corepressor 1), *IGFR1R* (insulin like growth factor 1 receptor) and *GNAS* (GNAS complex locus) were the most often mutated genes in patient-derived colon cancer cells (Figure 1E, left panel). MutSigCV analysis confirmed four recurrently mutated genes, named *APC* (APC regulator of WNT signaling pathway), *TP53* (tumor protein 53), *SUZ12* (polycomb repressive complex 2 subunit) and *ARID5B* (AT-rich interaction domain 5B), with critical statistical evidence (*p*-value < 0.2). In total, 29 distinct driver genes were discovered with these computational tools. *APC*, *SMAD4* (SMAD family member 4), *NF1* (neurofibromin 1), *ARID5B*, *NCOR1* (nuclear receptor co-repressor 1), *IGFR1R* (insulin like growth factor 1 receptor) and *GNAS* (GNAS complex locus) were the most often mutated genes in patient-derived colon cancer cells, YUMC-C1 and YUMC-C2 (Figure 1E, right panel). The tumor suppressor genes *APC* and *SMAD4* had an inordinately high number of frameshift or nonsense mutations. Moreover, the copy number modification in driver genes detected focal copy number variations (CNVs) at the exon level by CONTRA. The results proved that 12 driver genes had FCNV >20% in YUMC-C1 and YUMC-C2, including *GNAS*, *BRCA2* (DNA repair associated), *SOX9* (SRY-box transcription factor 9), *FGFR2* (fibroblast growth factor receptor 2), *SMAD4*, *SUZ12*, *FGFR3* (fibroblast growth factor receptor 3), *EGFR* (epidermal growth factor receptor), *EP300* (E1A binding protein p300), *NCOR1*, *TP53* and *IGF1R* (insulin like growth factor 1 receptor). Especially, GNAS presented a noticeable focal gain >50% of YUMC-C1 and YUMC-C2 compared with normal samples. Moreover, tumor suppressor genes, such as *TP53*, presented a noticeable focal loss >20%. The *EGFR* oncogene exhibited a noticeable focal gain >20%. A key regulator of cancer-associated fibroblasts, *TWIST1*, presented a noticeable focal gain >20%, and a regulator of angiogenesis, *VEGFB* (vascular endothelial growth factor B), presented a noticeable focal gain >40% (Figure 1F).

Taken together, we concluded that the study of drug-resistant and metastatic cancer cells could be of great value to therapeutic trials for the management of patients with drug-resistant properties.

### 2.2. Synergistic Anticancer Effect of Oxaliplatin and Lenvatinib on the Proliferation of Drug-Sensitive, Drug-Resistant, and Metastatic Colon Cancer Cells

According to a recent review, with regard to metastatic cancer, about 90% of therapeutic failure is a result of drug resistance [10]. We evaluated the synergistic anticancer effects of oxaliplatin and lenvatinib on drug-sensitive HT-29 colon cancer cells and patient-derived, drug-resistant and metastatic colon cancer cells. In addition, we carried out the viability and proliferation assay of YUMC-C1 and YUMC-C2 cells in the presence and absence of these compounds using MTT assays (Figure 2A–C). The viability and proliferation of HT-29 was significantly suppressed by either agent alone or the combination (Figure 2A). Notably, the viability and proliferation of the drug-resistant and metastatic YUMC-C1 and YUMC-C2 colon cancer cells, respectively, was significantly suppressed by the combination of both agents but not by either agent alone (Figure 2B,C). Immunoblot analyses of protein levels in HT-29, YUMC-C1 and YUMC-C2 cell lines indicated that the oxaliplatin and lenvatinib combination induced the most marked increases in the levels of p21, which are well-known arrestors of the cell cycle, and decreases in the levels of Ki-67, cyclin D1, CDK 4 and CDK 6, positive regulators of the proliferation and cell cycle, as compared with responses to either agent alone (Figure 2D). The anticancer drugs oxaliplatin and lenvatinib effectively suppressed the proliferation of drug-sensitive HT-29 colon cancer cells when administered alone and in combination (Figure 2A,D). On the other hand, however, only a combination of the oxaliplatin and lenvatinib was effectively suppressed in drug-resistant and metastatic YUMC-C1 and YUMC-C2 (Figure 2B–D). The half-maximal inhibitory concentration (IC_50_) of combination oxaliplatin and lenvatinib was the lowest for the HT-29, YUMC-C1, and YUMC-C2 cell groups (Table 1).

Collectively, these results suggest that the synergistic effect of oxaliplatin and lenvatinib looked to be potent against drug-resistant and metastatic colon cancer cells.

### 2.3. Drug-Resistant and Metastatic Colon Cancer Cells are Avoid from Therapeutic stress via EMT-Mediated Angiogenesis

The drug-sensitive HT-29 colon cancer cells were considerably affected by oxaliplatin and lenvatinib had no innate protective mechanisms. However, in contrast to the HT-29 cells, the drug-resistant and metastatic colon cancer cells apparently possessed mechanisms that enabled them to resist the lethal effects of the chemotherapeutic drug environment via EMT-mediated angiogenesis [23], which involves the growth of new blood vessels and is an essential component of the metastatic pathway (Figure 3A–E). Contrary to the HT-29 cells, YUMC-C1 and YUMC-C2, the drug-resistant and metastatic colon cancer cells, respectively, showed the significant secretion of VEGF following treatment with oxaliplatin and lenvatinib alone or in combination (Figure 3A–C). In addition, it is worth noting that E-cadherin expression had no significant difference, whereas increased EMT (vimentin, snail, and zeb1) expression in cytoplasm (Figure 3D) enhanced β-catenin nuclear translocation (Figure 3E) by either agent alone or the combination in YUMC-C1 and YUMC-C2. On the contrary, in HT-29, this led to the inhibition of vimentin, snail and zeb1 and dramatic increases in E-cadherin by either agent alone or the combination. This result has great implications for therapeutic trials.

### 2.4. Higher Levels of Stemness and Angiogenesis Markers and Metastatic Genes in Drug-Resistant and Metastatic Colon Cancer Cells than in Sensitive Anticancer Drug-Treated Cells

Considering that gene expression is largely influenced by environmental signals, we conjectured that drug-resistant or metastatic colon cancer cells would exhibit a stem-like phenotype with induced angiogenesis and metastatic genes by transcriptional reprogramming after treatment with anticancer drugs. To test this hypothesis, we conducted a genome-wide transcriptional profiling of drug-sensitive HT-29, drug-resistant and metastatic colon cancer cells, YUMC-C1 and YUMC-C2 (Figure 4). Numerous genes were notably differentially expressed between the HT-29, YUMC-C1, and YUMC-C2 cells, suggesting that multiple biological processes were reprogrammed in the drug-resistant and metastatic cancer cells but not in the drug-sensitive cells.

Interestingly, the results of the gene expression microarray analysis were in close agreement with our prediction. In the YUMC-C1 and YUMC-C2 drug-resistant and metastatic colon cancer cells, respectively, metastatic genes (*CDH2*, *KRAS*, *CDC42*, *MCPH1*, *SRGAP*) and markers of stemness (*BRACA1 and 2*, *CD44*, *KRT5*, *WNT4*, *CD29*, *SAL4*) were markedly enhanced (Figure 4). Especially, epithelial-mesenchymal transition (EMT)-related genes were more significantly induced in YUMC-C1 and YUMC-C2 cells than in HT-29 cells after treatment with oxaliplatin and lenvatinib either alone or in combination. This is according to well-known research or previous studies that the drug resistance of poorly differentiated cancer stem cells (CSCs) is related to EMT, which is mediated by the FGFR signaling pathway [24,25].

Collectively, these results proved that drug resistance and metastatic colon cancer cells were avoided to therapeutic trial and escape to harsh environment via transcriptional reprogramming involving EMT and angiogenesis regulatory pathways.

### 2.5. Combination Treatment with Oxaliplatin and Lenvatinib Reduced Tumor Size in Drug-Sensitive, Drug-Resistant, and Metastatic Colon Cancer Xenograft Models

We examined the synergistic anticancer effects of oxaliplatin and lenvatinib in vivo using a mouse xenograft tumor model induced using the patient-derived (drug-resistant and metastatic colon cancer cells, YUMC-C1 and YUMC-C2) and established (drug-sensitive HT-29) colon cancer cells. Similar to the results of the other experiments (Figure 2), oxaliplatin and lenvatinib, alone and in combination, significantly reduced tumor volumes and weights in the drug-sensitive HT-29 colon cancer xenograft models (Figure 5A,D). However, in the xenograft models of drug-resistant and metastatic colon cancer cells, YUMC-C1 and YUMC-C2, the agents did not significantly reduce tumor volumes and weights when administered alone, although their combination effectively did (Figure 5B,C,E,F). Moreover, no proof of treatment-related death or systemic toxicity was observed in any group, and there was also no significant change in the body weight of the mice treated with oxaliplatin and lenvatinib alone or in combination (Figure 5G–I). These results were some of the most notable discoveries, but, unfortunately, unlike normal colon cancer HT-29, drug resistance and metastatic colon cancer, YUMC-C1 and YUMC-C2 were prepared to escape a harsh environment via angiogenesis. The recruitment of new blood vessels is an essential component of the metastatic pathway when treated with oxaliplatin and lenvatinib with either agent alone or the combination (Figure 6A,B). Furthermore, we evaluated CD34, a marker of angiogenesis in cervical cancer, in HT-29, YUMC-C1, and YUMC-C2 cell xenograft tumors; the results demonstrated that angiogenesis activity was significantly induced in the drug-resistant and metastatic colon cancer tumor tissue treated with oxaliplatin and lenvatinib (Figure 6A,B).

The synergistic anti-cancer effect of oxaliplatin and lenvatinib was considered to be potent in a therapeutic trial of drug resistance and metastatic colon cancer cells; however, in the absence of conclusive evidence, we should err on the side of caution.

## 3. Discussion

Resistance to cancer chemotherapy is a major cause of concern. The clinical characteristics of drug-resistant cancer are noticeably different in terms of genetic and molecular profiles [26,27,28]. Therefore, to better understand the differences between sensitive and resistant colon cancer cells, we compared HT-29 cells (sensitive) with YUMC-C1 and YUMC-C2 cells (patient-derived resistant and metastatic) after treatment with lenvatinib, oxaliplatin, and their combination both in vitro and in vivo. The patient-derived cell lines that we used were acquired from a recurrent and metastasis case after administered neoadjuvant FOLFOX. Of the two kinds of patient-derived cell lines, only oxaliplatin may have been shown to have resistance. Lenvatinib was used because it is a type of tyrosine kinase inhibitor (TKI) as one of the second-generation cancer treatments, unlike conventional chemotherapy, and is known to have a strong anticancer activity among TKIs. In addition, if there is an additional opportunity, the next study will also confirm its relationship to angiogenesis. We found that the combination of lenvatinib and oxaliplatin inhibited the proliferation of colon cancer cells (in vitro) and led to tumor shrinkage (in vivo) to a greater extent than treatment with either agent alone. Oxaliplatin is a well-known chemotherapeutic drug that kills various cancer cells by forming DNA lesions and inhibiting RNA and DNA synthesis [29,30,31]. Lenvatinib is a multitargeted tyrosine kinase inhibitor with significant activity on various solid tumors [32,33,34].

A remarkable number of patients were identified as having an uncharted genetic event, indicating that other genetic or epigenetic factors may be associated with the pathogenesis and biological behavior of drug-resistant cancer [35,36,37]. DNA microarray findings suggest that proteomics analysis and targeted therapy are focus areas to overcome drug resistance. Although new approaches using contemporary chemotherapeutic agents are growing rapidly, effective agents against drug-resistant cancer have not yet been discovered. In addition, drug-resistant cancer cells are highly likely to recur and metastasize, causing death [38,39,40]. Cancer cell resistance to anticancer drugs can also be acquired via individual genetic differences, particularly in tumoral somatic cells [41,42]. Other cancer drug resistance mechanisms include multidrug resistance, the modification of drug metabolism, the suppression of cancer cell death, and the epigenetic alteration of drug targets [43,44,45]. The reduction of tumor size in patients can be considered a partial or complete response, and such a response to chemotherapy is favored by most clinicians. However, the outcome of chemotherapy should also be studied from a molecular perspective.

The two tissue samples used in this study were from patients diagnosed with colon cancer liver metastasis 7 and 2 years before the study. Both patients underwent effective preoperative chemotherapy. However, after surgery, there was persistent disease recurrence, and chemotherapy became ineffective. After several lines of chemotherapy, cancer cells can continue to acquire molecular-based drug resistance. The combination of oxaliplatin and lenvatinib is known to be effective in suppressing cancer cell proliferation. However, this effect was not observed for the patient-derived drug-resistant colon cancer cells in this study.

EMT-mediated angiogenesis allows cancer cells to evade anticancer drug effects [23]. Following this, the drug-resistant colon cancer cells probably evaded the combination effects of oxaliplatin and lenvatinib via the induction of angiogenesis. Microarray analysis indicated that the drug-resistant colon cancer cells expressed considerably higher levels of genes related to metastasis, stemness, and angiogenesis than the drug-sensitive colon cancer cells when treated with oxaliplatin and lenvatinib alone or in combination.

Our study results should be interpreted considering the limitation that cells were obtained from only two patients. However, studying the genetic change of drug resistance and metastatic cancer when exposed an anti-cancer drug, it would be possible to suggest an appropriate clinical solution to a recurrent and metastatic cancer patient.

Tumor shrinkage alone cannot be considered the sole factor deciding treatment outcome because the remaining cancer cells can become resistant to the anticancer treatment and lead to a relapse. We demonstrated that lenvatinib and oxaliplatin alone or in combination induced higher expression levels of EMT-related genes in YUMC-C1 and YUMC-C2 cells than in HT-29 cells. This finding is in agreement with the results of previous studies that the drug resistance of poorly differentiated cancer stem cells is related to EMT, which occurs through the fibroblast growth factor receptor signaling pathway [24,25].

Collectively, our results indicate that the drug resistance and metastasis of colon cancer cells are mediated by transcriptional reprogramming involving EMT and angiogenesis regulatory pathways. Although the combination of oxaliplatin and lenvatinib was effective, further research is needed to obtain conclusive evidence.

## 4. Materials and Methods

### 4.1. Characteristics of the Sample-Donating Patients

#### 4.1.1. Patient 1

Patient 1 was a 71-year-old woman with sigmoid colon cancer with liver metastasis who received neoadjuvant folinic acid, oxaliplatin, and fluorouracil (FOLFOX) chemotherapy 7 years prior and underwent anterior resection with wedge resection of the liver. She also underwent four additional wedge liver resections and two intraoperative liver radiofrequency ablations. The specimens for culture were obtained after the last operation. After wedge liver resection (segment 8), the pathology report indicated the presence of metastatic adenocarcinoma.

#### 4.1.2. Patient 2

Patient 2 was a 64-year-old man who was administered neoadjuvant FOLFOX with cetuximab and underwent a low anterior resection with wedge liver resection surgery. Two years later, multiple peritoneal masses and a metastatic lesion were found in the liver, and additional surgery was performed. Specimens for culture were obtained from the masses in the liver, and metastatic adenocarcinoma was confirmed in the pathology report.

### 4.2. Patient Tissue Specimens

Fresh tumors were acquired from patients with biochemically and histologically established colon cancer with liver metastasis who were treated at the Severance Hospital, Yonsei University College of Medicine, Seoul, Korea. Fresh tumors were obtained during surgical resection of the metachronous metastatic site in the liver.

### 4.3. Ethical Considerations

The research protocol was approved by the Institutional Review Board of Severance Hospital, Yonsei University College of Medicine (IRB Protocol: 3-2019-0281). Cell samples were acquired from patients at the Severance Hospital, Yonsei University College of Medicine, Seoul, Korea). Cell samples were acquired from patients at the Gangnam Severance Hospital, Yonsei University College of Medicine, Seoul, Korea.

### 4.4. Tumor Cell Isolation and Primary Culture

After resection, tumor tissue samples were maintained in normal saline supplemented with antifungal and antibiotic agents and transferred to the laboratory. Normal tissue and fat were removed, and the tumor tissues were rinsed with 1× Hank’s Balanced Salt Solution. Further protocol details are described in our previous article [46].

### 4.5. Cell Culture

The patient-derived colon cancer and HT-29 cells (ATCC, Manassas, VA, USA) were grown in Roswell Park Memorial Institute-1640 medium with 10–15% fetal bovine serum. The cells were authenticated using short tandem repeat profiling, karyotyping, and isoenzyme analysis. Mycoplasma contamination was invariably checked with the Lookout Mycoplasma PCR Detection Kit (Sigma-Aldrich; MP0035).

### 4.6. Preparation of DNA

FFPE DNAs were extracted with the QIAamp DNA FFPE Tissue Kit (Qiagen, Valencia, CA, USA), pursuant to the manufacturers’ manuals. Initial QC checks of FFPE DNA were performed with electrophoresis on 1% agarose gels and the Qubit dsDNA HS Assay Kit with the Qubit 2.0 fluorometer (Life Technologies, Carlsbad, CA, USA) pursuant to manufacturers’ manuals.

### 4.7. Preparation of Libraries

Libraries were arranged with the SureSelect XT protocol (Agilent Technologies, Santa Clara, CA, USA) with Custom Panel by the Macrogen (Macrogen, Seoul, Korea). Their quality checked with the 2100 Bioanalyzer (Agilent). A size of the product of 200~400 bp was craved. Then, the libraries were then quantified with the Qubit dsDNA HS Assay Kit and the Qubit 2.0 fluorometer (Life Technologies). The libraries were sequenced paired end (2 x 150 bp) on a NextSeq500 instrument (Illumina, San Diego, CA, USA) with high output with Sequencing by Synthesis chemistry.

### 4.8. Analysis of DNA Sequences

The adapter sequences were eliminated by fastp (Chen, 2018). Trimmed reads were aligned to the reference genome (GRCh37/hg19) with BWA-MEM (Li, 2013). Poorly mapped reads that have mapping quality (MAPQ) below 20 were eliminated with Samtools version 1.3.1 (Li et al., 2009). Duplicated reads were removed with Sambamba markdup (version 0.6.7) (Tarasov, 2015). Base quality of deduplicated reads was recalibrated with GATK BaseRecalibrator. Somatic mutations including single nucleotide variants (SNVs), small insertions and deletions (INDELs) were identified with the MuTect2 algorithm (Cibulskis et al., 2013). Then, false-positive variant calls that originated from the oxoG artifact were eliminated. Moreover, mutations below 2 % variant allele frequency (VAF) and 100X total depth were eliminated. Variants were eliminated when minor allele frequency (MAF) ≥ 1% in genomAD, ExAC and Macrogen Korean Population Database. All the remained variants were annotated with SnpEff & SnpSift v4.3i (Cingolani et al., 2012a; Cingolani et al., 2012b).

### 4.9. Copy Number Variation Analysis

Focal copy number variations (CNVs) were detected with CONTRA v2.0.8 software between YUMC-C1/C2 and paired normal samples with default parameters, except for ‘-removeDups’. Focal CNVs were detected between deduplicated bam files of YUMC-C1/C2 and the merged bam file of eight normal samples. Focal CNVs with *p*-values < 0.05 were considered statistically significant. The fraction of CNV (FCNV) was computed as follows: FCNV = (size of significant CNVs)/size of exons. Details were pursuant to the manufacturers’ manuals.

### 4.10. Cell Viability Assay

Cell proliferation was measured using the 3-(4,5-dimethylthiazol-2-yl)-5-(3- carboxymethoxyphenyl)-2-(4-sulfophenyl)-2H-tetrazolium (MTT) assay (Roche, Basel, Switzerland; 11465007001). Cells were seeded in 96-well plates at 6 × 10^3^ cells per well and incubated overnight to achieve 90% confluency. The indicated drugs were added to achieve final concentrations of 0–400 μM. Cells were then incubated for the indicated durations of time before cell viability was determined using the MTT reagent according to the manufacturer’s protocol. Absorbance was measured at 550 nm. Viable cells were counted using trypan blue exclusion. Data were calculated as a percentage of the signal observed in vehicle-treated cells; the results are shown as the mean ± standard error of the mean of triplicate experiments.

### 4.11. Enzyme-linked Immunosorbent Assay (ELISA)

Human vascular endothelial growth factor (VEGF) protein level in the cultured cells (26–36 h) was determined using the Quantikine ELISA kit (R&D Systems, Minneapolis, MN, USA). Volumes of cell culture supernatants were normalized to the total number of cells present at the time of collection. For each experiment, triplicate samples were measured for statistical significance. This assay employs an antibody specific to Human VEGF coated on a 96-well plate, and biotinylated anti-human VEGF antibody is added. After washing away unbound biotinylated antibody, HRP-conjugated streptavidin is pipetted to the wells.

### 4.12. Immunofluorescence Analysis and Confocal Imaging

The expression of β-catenin was analyzed by immunofluorescence staining. Cells cultured on glass-bottomed dishes (MatTek, Ashland, MA. USA) were fixed with 4% formaldehyde solution (R&D Systems, Abingdon, UK) for 10 min and permeabilized with 0.5% TritonX-100 in phosphate-buffered saline (PBS) for 10 min. Slides were air dried, washed with PBS, and incubated with anti-β-catenin (1:25; Abcam, Cambridge, UK) in 3% bovine serum albumin in PBS. After being washed with PBS, slides were incubated with Alexa 488 (1:200; Molecular Probes, Eugene, OR, USA). Nuclei were stained with Hoechst 33342 (Life Technologies, Grand Island, NY, USA) for visualization. Images were observed under a confocal microscope (LSM Meta 700; Zeiss, Oberkochen, Germany) and were analyzed using the Zeiss LSM Image Browser, version 4.2.0121.

### 4.13. Microarray Experiment and Data Analysis

RNA purity and integrity were evaluated using an ND-1000 spectrophotometer (NanoDrop, Wilmington, DE, USA) and Agilent 2100 bioanalyzer (Agilent Technologies, Palo Alto, CA, USA). RNA labeling and hybridization were performed using the Agilent One-Color Microarray-based Gene Expression Analysis protocol (Agilent Technologies, V 6.5, 2010). Additional protocol and data analysis details are described in our previous article [47].

### 4.14. Immunoblot Analysis

Cells were washed twice with cold PBS and lysed on ice using a protein extraction buffer (Pro-Prep, iNtRON Biotechnology, Seoul, Korea) following the manufacturer’s protocol. Protein concentrations were determined using a bicinchoninic acid assay (Pierce Biotechnology, Rockford, IL, USA); equal amounts of protein (20 μg) were separated by 8–10% sodium dodecyl sulfate-polyacrylamide gel electrophoresis. They were then electro-transferred onto polyvinylidene fluoride membranes (Millipore, Bedford, MA, USA). The membranes were subsequently blocked with 5% nonfat milk in Tris-buffered saline plus Tween (TBST) for 1 h at room temperature and incubated with appropriate concentrations of primary antibodies against Ki-67, p21, vimentin, E-cadherin, Snail, and Zeb1 obtained from Abcam (Cambridge, UK). The immunoblots were incubated with cyclin-dependent kinase (CDK) 4, CDK 6, cyclin D1, and β-actin antibodies (Santa Cruz Biotechnology, Santa Cruz, CA, USA) overnight at 4 °C. The membranes were then rinsed three to five times with TBST and probed with the corresponding secondary antibodies conjugated to horseradish peroxidase (Santa Cruz) at room temperature for 1 h. After rinsing, the blots were developed using enhanced chemiluminescence reagents (Pierce) and exposed using Kodak X-OMAT AR film (Eastman Kodak, Rochester, NY, USA) for 3–5 min.

### 4.15. Immunohistochemistry

All tissues were fixed in 10% neutral-buffered formalin and embedded in paraffin wax following standard protocols. Tissue sections (5 μm) were dewaxed, and antigen retrieval was performed in citrate buffer (pH 6) using an electric pressure cooker set at 120 °C for 5 min. Sections were incubated for 5 min in 3% hydrogen peroxide to quench the endogenous tissue peroxidase; then, they were incubated with primary monoclonal antibodies against CD-34 (Abcam) diluted with PBS (1:100) overnight at 4 °C. All tissue sections were counterstained with hematoxylin, dehydrated, and then mounted.

### 4.16. Image Analysis

The MetaMorph 4.6 software (Universal Imaging Co., Downingtown, PA, USA) was used for the computerized quantification of immunostained target proteins.

### 4.17. Human Colon Cancer Cell Xenograft Mouse Model

HT-29, YUMC-C1 and YUMC-C2 human colon cancer cells (4.5 × 10^6^ cells/mouse) were cultured in vitro and then injected subcutaneously into the upper left flank region of 6-week-old female BALB/c nude and NOD/Shi-scid, IL-2Rγ KOJic (NOG) mice. After 15 days, when the tumor size reached approximately 100–200 mm^3^, tumor-bearing mice were randomly grouped (*n* = 10/group) to receive 17 mg/kg oxaliplatin (i.p.), 10 mg/kg lenvatinib p.o.), or a combination of 8.5 mg/kg oxaliplatin (i.p.) and 4.5 mg/kg lenvatinib (p.o.) once every 2 days. Tumor size was measured every other day using calipers; tumor volume was then estimated using the following formula: L × S^2^/2 (where L and S are the longest and shortest diameters, respectively). Animals were maintained under specific pathogen-free conditions, and all experiments were approved by the Animal Experiment Committee of Yonsei University.

### 4.18. Statistical Analysis

Statistical analyses were performed using GraphPad Prism 6.0 (GraphPad Software Inc., La Jolla, CA, USA), and the immunohistochemistry results were evaluated via analysis of variance followed by the Bonferroni *post hoc* test. Values are expressed as the mean ± standard deviation (SD), and *p* < 0.05 indicated statistical significance.

## 5. Conclusions

Multidrug-resistance-related genetic alternation in patient-derived, multidrug-resistant colon cancers was observed after the treatment of oxaliplatin and lenvatinib. Consequentially, these findings can be useful to design future rational clinical studies on patients with recurrent or metastatic colon cancer cells in order to develop effective therapies.

## Figures and Tables

**Figure 1 ijms-21-07469-f001:**
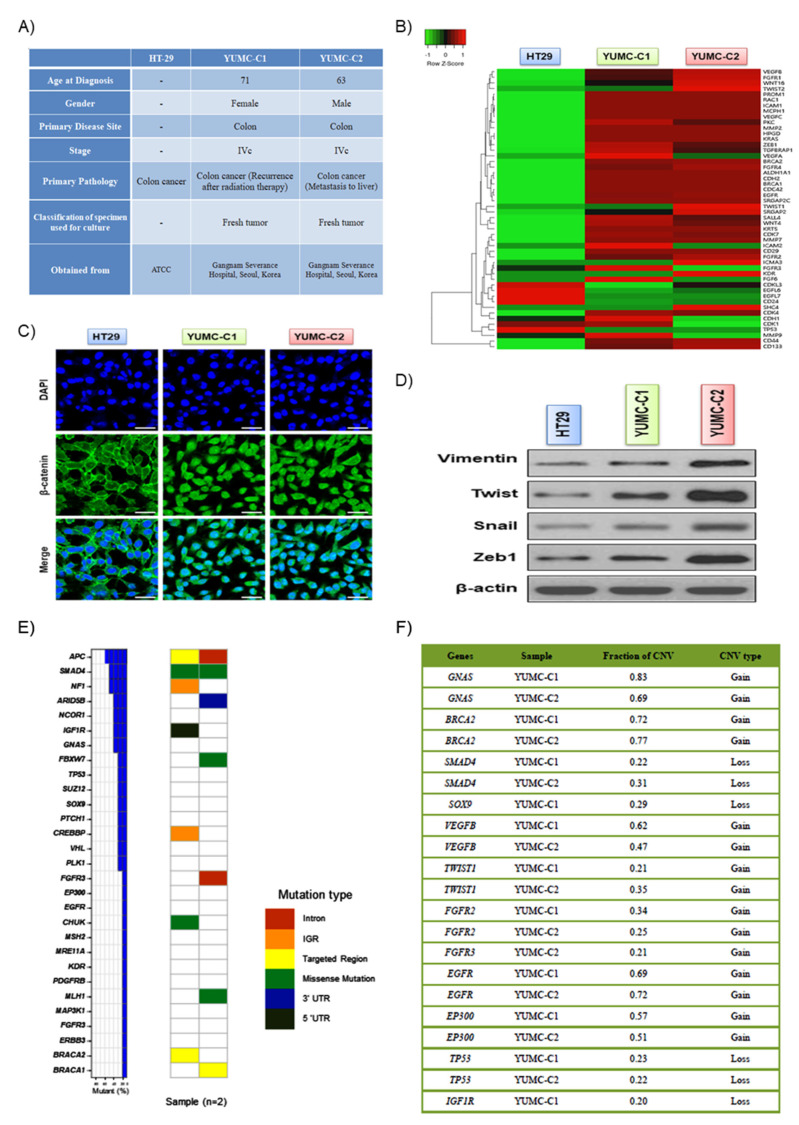
(**A**) Cell line characteristics, viability after drug treatment of all colon cancer cell lines examined. (**B**) Gene expression profiles of colon cancer cells. Gene expression analysis of HT-29 (American Type Culture Collection, ATCC) and YUMC-C1 and YUMC-C2 patient-derived colon cancer cells. (**C**) Immunofluorescence cytochemical staining about β-catenin in HT-29, YUMC-C1, and YUMC-C2 cells examined at 400× magnification; scale bar, 20 μm. (**D**) Immunoblot about markers of epithelial-mesenchymal transition (EMT). (**E**) Mutation rates of 29 driver genes in YUMC-C1 and YUMC-C2. The right panel represents the distribution of all mutations, which are colored pursuant to the mutation type. (**F**) Fraction of significant CNVs of driver genes in patient-derived colon cancer cells. All experiments were repeated at least 3 times.

**Figure 2 ijms-21-07469-f002:**
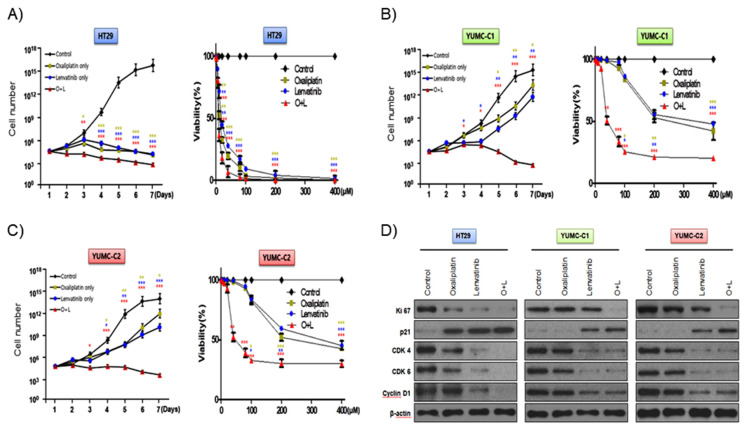
Synergistic anticancer effects of oxaliplatin and lenvatinib on HT-29 cells from American Type Culture Collection (ATCC) and YUMC-C1 and YUMC-C2 patient-derived colon cancer cells. Proliferation assay of (**A**) HT-29, (**B**) YUMC-C1, (**C**) YUMC-C2 cells after treatment with oxaliplatin and lenvatinib alone or in combination. (**D**) Immunoblot analysis of related cell cycle proteins in HT-29, YUMC-C1, and YUMC-C2 cells. Data points are the means (%) of values of the solvent-treated control. All experiments were repeated at least three times. Data are the means ± SD. * *p* < 0.05, ** *p* < 0.01, and *** *p* < 0.005 compared with the control.

**Figure 3 ijms-21-07469-f003:**
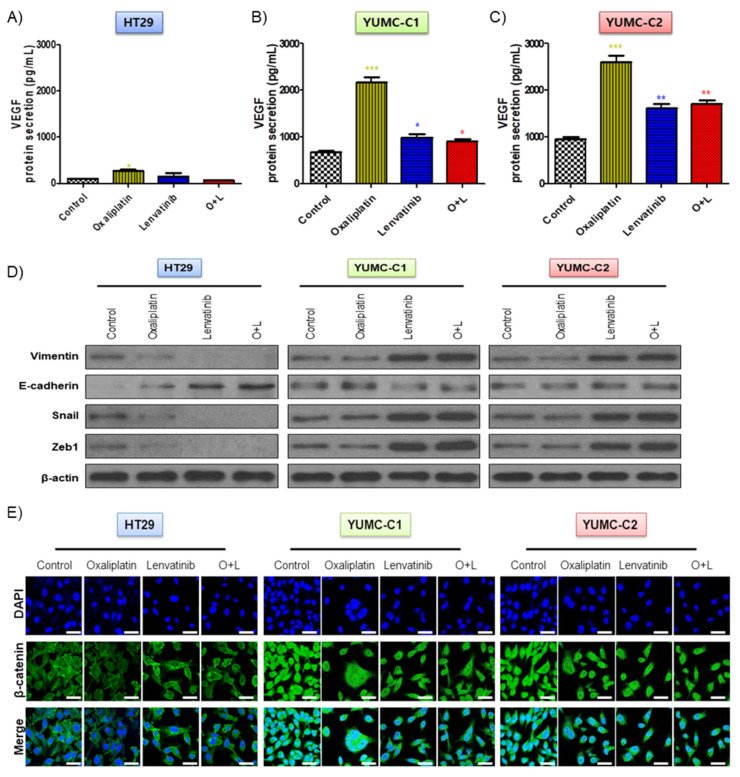
(**A**–**C**) Enzyme-linked immunosorbent assay (ELISA) of secreted vascular epithelial growth factor (VEGF) expression in conditioned media of HT-29, YUMC-C1, and YUMC-C2. (**D**) Immunoblot analysis and. (**E**) Examined at 400× magnification; scale bar, 20 μm, immunofluorescence cytochemical staining about β-catenin, markers of EMT in HT-29, YUMC-C1, and YUMC-C2 cells. VEGF secretion and EMT expression were suppressed to a greater extent in HT-29 cells than in YUMC-C1 and YUMC-C2 cells. All experiments were repeated at least three times. Data are the means ± SD. * *p* < 0.05, ** *p* < 0.01, and *** *p* < 0.005 compared with the control.

**Figure 4 ijms-21-07469-f004:**
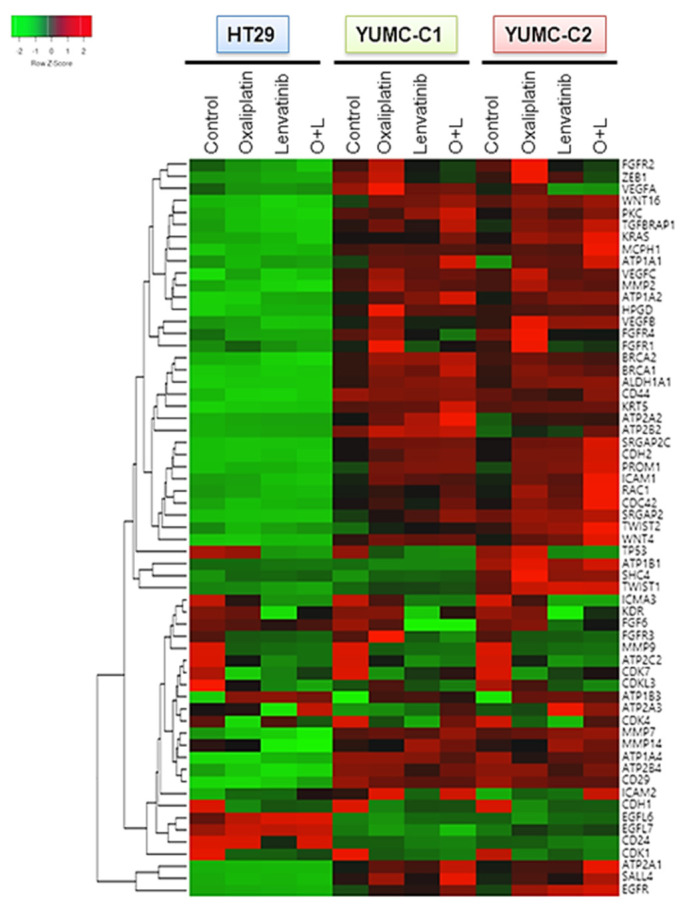
Gene expression profiles of colon cancer cells. Gene expression analysis of HT-29 (American Type Culture Collection, ATCC) and YUMC-C1 and YUMC-C2 patient-derived colon cancer cells treated with oxaliplatin and N-hydroxy-7-(2-naphthalenylthio)-heptanamide (lenvatinib) alone or in combination using a microarray approach. Hierarchical clustering analysis of a comparison of HT-29, YUMC-C1, and YUMC-C2 cell samples.

**Figure 5 ijms-21-07469-f005:**
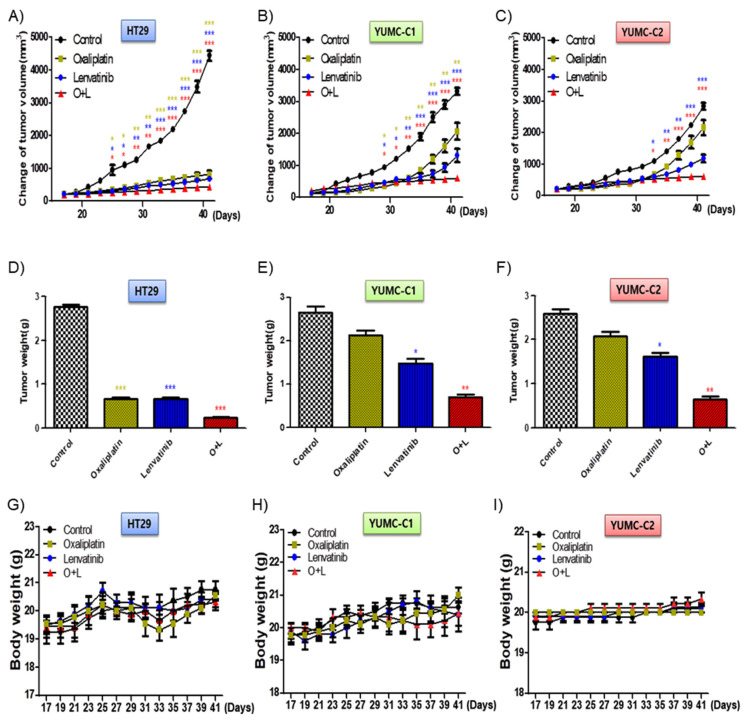
Cotreatment with oxaliplatin and lenvatinib induced a higher tumor shrinkage of cancer cell xenografts in vivo and inhibited tumor progression to a greater extent than treatment with either agent alone. Athymic nude mice with established tumors were treated with the indicated inhibitors. Data are the mean tumor volumes. The inhibition of tumor progression of HT-29 (**A**,**D**,**G**), YUMC-C1 (**B**,**E**,**H**), and YUMC-C2 (**C**,**F**,**I**) cell-induced tumors treated with agents alone or in combination. (**A**–**C**) Change in tumor volume, (**D**–**F**) weight of dissected tumors, and (**G**–**I**) compounds had no significant effect on mouse body weight. * *p* < 0.05, ** *p* < 0.01, and *** *p* < 0.005 compared with control.

**Figure 6 ijms-21-07469-f006:**
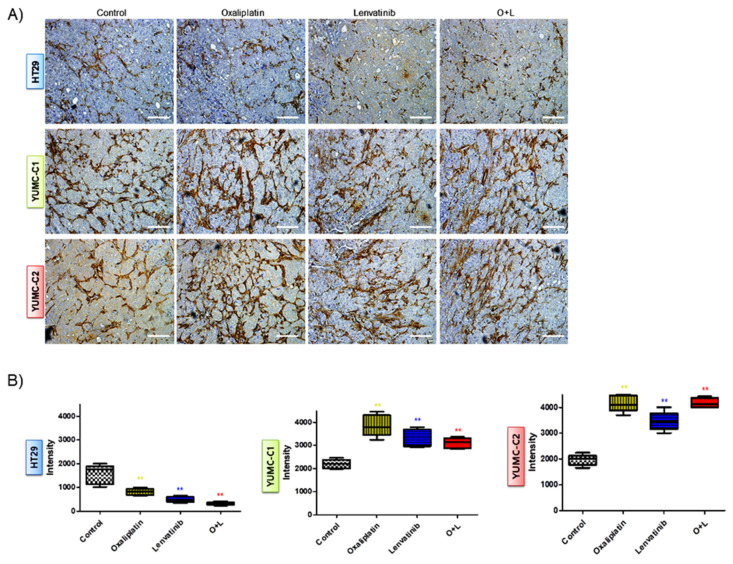
(**A**) Immunohistochemical analysis of CD34 protein levels in paraffin-embedded xenograft tumor tissues. Examined at 400× magnification; scale bar, 80 μm. Assay estimated marker levels of angiogenesis-related protein (CD34) in HT-29, YUMC-C1, and YUMC-C2 cell samples. (**B**) Image Analysis, MetaMorph 4.6 image-analysis software was used to quantify CD-34 immunostaining. ** *p* < 0.01 compared with control.

**Table 1 ijms-21-07469-t001:** IC_50_ (half maximal inhibitory concentration) determination using a cell proliferation assay. Oxaliplatin and lenvatinib combination treatment is a lower IC_50_ than oxaliplatin and lenvatinib alone. Each data point represents the mean of 3 independent MTT assays for IC_50_ performed in triplicate. SD, standard deviation. * *p* < 0.05.

	Hisopathology	Animal	Cell Proliferation IC_50_^*^ (μM)		
			**Oxaliplatin**	**Lenvatinib**	**Oxaliplatin** + **Lenvatinib**
**HT-29**	Colon cancer	Human	2.2 (±0.3)	14.5 (±0.3)	0.9 (±0.2) + 7.9 (±0.3)
**YUMC-C1**	Colon cancer	Human	241.2 (±0.1)	240.8 (±0.2)	50.3 (±0.2) + 61.4 (±0.3)
**YUMC-C2**	Colon cancer	Human	247.1 (±0.3)	247.5 (±0.1)	53.4 (±0.1) + 63.2 (±0.2)

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
