# Peer review of "Patient-Derived, Drug-Resistant Colon Cancer Cells Evade Chemotherapeutic Drug Effects via the Induction of Epithelial-Mesenchymal Transition-Mediated Angiogenesis"

_ijms, 2020, doi:10.3390/ijms21207469_

Round 1

Reviewer 1 Report

The study is very interesting. I suggest few minor language and editing corrections. 

Point 1: line 47. The first sentence is too long and need to be rearrange.

Point 2: line 57. Change-sufficient to only identify 

Point 3: line 81. Especially,  Nuclear  nuclear. Whole sentence needs to be change.

Point 4. In Fig. 1 is missing the number of repeated experiment.

Point 5. Have the authors checked all reagents by Limulus amebocyte lysate assay because your cells could present more active phenotype ? 

Point 6. Did you check that your data are parametric and followed the gaussian distribution to used ANOVA test?

Point 7. Are your cell lines used in experiments mycoplasma free.

Author Response

Reviewer 1

  1. line 47. The first sentence is too long and need to be rearrange.

> Reply: I don't know how to thank you enough. I have made the suggested correction, line 47.

  1. line 57. Change-sufficient to only identify

> Reply: Thank you for your comment. I have made this correction, line 57.

  1. line 81. Especially, Nuclear, nuclear. Whole sentence needs to be change.

> Reply: I have made this correction, line 81.

  1. . In Fig. 1 is missing the number of repeated experiment.

> Reply: Thank you for your comment. I have added ‘the number of repeated experiment’

  1. Have the authors checked all reagents by Limulus amebocyte lysate assay because your cells could present more active phenotype?

> Reply: Yes, of course. Your question is correct.

  1. Did you check that your data are parametric and followed the gaussian distribution to used ANOVA test?

> Reply: Yes, of course. I did as you said.

  1. Are your cell lines used in experiments mycoplasma free.

> Reply: Thank you for your comment. Yes, of course. I have added sentence as you said, line 337-338.

Reviewer 2 Report

In this manuscript, the authors compared between HT-29 and patient-derived multi drug resistant colon cancer cells on the effect after the treatment of oxaliplatin and lenvatinib. More importantly, the authors did a great job in analyzing the genetic alternation before and after treatment in both sensitive and resistant cell lines. I think the data presented has a good quality. However, the discussion and statement presented can be improved and reconsidered before acceptance. 

The main concern is that I feel the title and conclusion on line 437 is not supported. We did see the synergistic cell-growth suppression effect of oxaliplatin and lenvatinib in HT-29, YUMC-C1 and YUMC-C2 cells. However, from the in vitro or in vivo data, I didn't cells evading the anticancer drug effects. Two patients have never been given the co-treatment of oxaliplatin and lenvetinib. So it is in a lack of evidence to demonstrate the cancer cells can evade therapeutic drug. In addition, the authors focused on the gene expression on EMT-related genes and showed the over expression of these genes after treatment of oxaliplation and lenvatinib. Some other critical genes are missed. For example, the TP53, FGFR are suppressed after treatment as shown in Figure 4. Do they mean treatment can also induce resistance through less cell cycle arrest and less fibroblast covering? It would be biased if only demonstrating the cells evade multi drug resistance by angiogenesis. Actually from the data, I can see more genetic factors that involves. Thus, I would like to discuss and hear back from the authors if they think changing the main conclusion to "Multridrug resistance-related genetic alternation in patient-derived multi drug resistant colon cancers after treatment of oxaliplatin and lenvatinib". I think it will still be very interesting to readers and meet the data presented. 

My second concern is that the discussion section can be flourished. For example, the authors did a great job in analyzing different genetic targets. However, in line 279-284, it's only briefly mentioned that higher levels of genes related with metastasis, stemness and angiogenesis. How about other genes the authors has tested? Similarly, on line 198, please specify the "metastatic genes and markers of stemless were markedly enhanced". What are the exact genes the authors are referring to. I would like to see more discussion on the data in Figure 4, which includes significant amount of information, even though only 2 patient sample were derived.

Other minor concerns include: 

1. Please consider rephrase line 63, some genetic mechanisms have been reported and a bunch of molecular targets have been identified, such as survivin, bcl-2, MMR, NER systems and so on. I think the author means the exact or detailed mechanisms prevents the drug-induced apoptosis is unknown. 

2. What's the unit in Table 1

3. The abbreviation used throughout the text should be modified. Abbreviation should be explained the first time they appear in the context. For example, EMT first appears on line 82, but explained on line 84; APC first appears on line 84 but explained on line 88. There are multiple problems like this, which cause confusion to readers. I did see some abbreviation listed at the end of the manuscript. But they can be place in the beginning and more can be included.

4. In Figure 2D, why author decided to check on cell cycle proteins. Especially since Lenvatinib inhibit EGFR and oxaliplatin forms DNA adducts, what's the thought behind. In addition, what's the rationality of choosing the combination of oxaliplatin and lenvatinib to investigate the evasion of multi drug resistance. Why not using other types of chemotherapeutics, such as paclitaxel or 5-FU. I couldn't find these information in Introduction or Discussion. 

5. On line 163, 164, I suggest delete "angiogenesis stimulation induced by" because from Figure 3A-C, only overexpression of VEGF can be interpreted but not the angiogenesis stimulation. 

6. On line 285, it's good that the authors mentioned only two patient samples were derived. But then what? Just a sentence there without follow-up illustration is weird.

7. Please briefly expand section 4.11. How proteins are extracted from cells? What are the primary and secondary antibodies used.  

Author Response

Reviewer 2

  1. In this manuscript, the authors compared between HT-29 and patient-derived multi drug resistant colon cancer cells on the effect after the treatment of oxaliplatin and lenvatinib. More importantly, the authors did a great job in analyzing the genetic alternation before and after treatment in both sensitive and resistant cell lines. I think the data presented has a good quality. However, the discussion and statement presented can be improved and reconsidered before acceptance.

> Reply: I don't know how to thank you enough.

  1. The main concern is that I feel the title and conclusion on line 437 is not supported. We did see the synergistic cell-growth suppression effect of oxaliplatin and lenvatinib in HT-29, YUMC-C1 and YUMC-C2 cells. However, from the in vitro or in vivo data, I didn't cells evading the anticancer drug effects. Two patients have never been given the co-treatment of oxaliplatin and lenvetinib. So it is in a lack of evidence to demonstrate the cancer cells can evade therapeutic drug. In addition, the authors focused on the gene expression on EMT-related genes and showed the over expression of these genes after treatment of oxaliplation and lenvatinib. Some other critical genes are missed. For example, the TP53, FGFR are suppressed after treatment as shown in Figure 4. Do they mean treatment can also induce resistance through less cell cycle arrest and less fibroblast covering? It would be biased if only demonstrating the cells evade multi drug resistance by angiogenesis. Actually from the data, I can see more genetic factors that involves. Thus, I would like to discuss and hear back from the authors if they think changing the main conclusion to "Multridrug resistance-related genetic alternation in patient-derived multi drug resistant colon cancers after treatment of oxaliplatin and lenvatinib". I think it will still be very interesting to readers and meet the data presented.

> Reply: Thank you for your suggestion. I agree with you and changed the main conclusion as you suggested, line 451-452

  1. My second concern is that the discussion section can be flourished. For example, the authors did a great job in analyzing different genetic targets. However, in line 279-284, it's only briefly mentioned that higher levels of genes related with metastasis, stemness and angiogenesis. How about other genes the authors has tested? Similarly, on line 198, please specify the "metastatic genes and markers of stemless were markedly enhanced".

> Reply: I have added name of genes, line 199~200.

What are the exact genes the authors are referring to. I would like to see more discussion on the data in Figure 4, which includes significant amount of information, even though only 2 patient sample were derived.

> Reply: Thank you for your comments. I have added sentence on discussion as you suggested, line 287-290.

Other minor concerns include:

  1. Please consider rephrase line 63, some genetic mechanisms have been reported and a bunch of molecular targets have been identified, such as survivin, bcl-2, MMR, NER systems and so on. I think the author means the exact or detailed mechanisms prevents the drug-induced apoptosis is unknown.

> Reply: I have corrected everything you have pointed out, line 64-66.

  1. What's the unit in Table 1

> Reply: I have corrected everything you have pointed out.

  1. The abbreviation used throughout the text should be modified. Abbreviation should be explained the first time they appear in the context. For example, EMT first appears on line 82, but explained on line 84; APC first appears on line 84 but explained on line 88. There are multiple problems like this, which cause confusion to readers. I did see some abbreviation listed at the end of the manuscript. But they can be place in the beginning and more can be included.

> Reply: Thank you for your comments. I have corrected everything you have pointed out.

  1. In Figure 2D, why author decided to check on cell cycle proteins. Especially since Lenvatinib inhibit EGFR and oxaliplatin forms DNA adducts, what's the thought behind. In addition, what's the rationality of choosing the combination of oxaliplatin and lenvatinib to investigate the evasion of multi drug resistance. Why not using other types of chemotherapeutics, such as paclitaxel or 5-FU. I couldn't find these information in Introduction or Discussion.

> Reply: Thank you for your comments. I have corrected everything you have pointed out in discussion, line 260-265. The patients derived cell lines what we used were acquired from recurrent and metastasis case after administered neoadjuvant FOLFOX. As you mentioned it, the reason why 5-FU was not used is because of resistance of FOLFOX chemotherapy, and since oxaliplatin is stronger than 5-FU, it is thought that only oxaliplatin may have been shown to have resistance. Lenvatinib was used because it is a type of tyrosine kinase inhibitor (TKI) as one of the second generation cancer treatments, unlike conventional chemotherapy, and is known to have a strong anticancer activity among TKIs. In addition, if there is an additional opportunity, the next study will also confirm its relationship to angiogenesis.

  1. On line 163, 164, I suggest delete "angiogenesis stimulation induced by" because from Figure 3A-C, only overexpression of VEGF can be interpreted but not the angiogenesis stimulation.

> Reply: I have corrected everything you have pointed out.

  1. On line 285, it's good that the authors mentioned only two patient samples were derived. But then what? Just a sentence there without follow-up illustration is weird.

> Reply: Thank you for your comments. I have added sentence on discussion as you suggested, line 287-290.

  1. Please briefly expand section 4.11. How proteins are extracted from cells? What are the primary and secondary antibodies used.

> Reply: I have corrected everything you have pointed out, line 380-385
